# Broad and colossal edge supercurrent in Dirac semimetal Cd$_3$As$_2$ Josephson junctions

Chun-Guang Chu [1], Jing-Jing Chen [2,3], An-Qi Wang [1]✉,
Zhen-Bing Tan [2,3]✉, Cai-Zhen Li[2,3], Chuan Li [4], Alexander Brinkman[4],
Peng-Zhan Xiang[1], Na Li[1], Zhen-Cun Pan[1], Hai-Zhou Lu[2], Dapeng Yu [2,3,5] &
Zhi-Min Liao [1,5]✉

Edge supercurrent has attracted great interest recently due to its crucial role in achieving and manipulating topological superconducting states. Proximity-induced superconductivity has been realized in quantum Hall and quantum spin Hall edge states, as well as in higher-order topological hinge states. Non-Hermitian skin effect, the aggregation of non-Bloch eigenstates at open boundaries, promises an abnormal edge channel. Here we report the observation of broad edge supercurrent in Dirac semimetal Cd$_3$As$_2$-based Josephson junctions. The as-grown Cd$_3$As$_2$ nanoplates are electron-doped by intrinsic defects, which enhance the non-Hermitian perturbations. The superconducting quantum interference indicates edge supercurrent with a width of ~1.6 μm and a magnitude of ~1 μA at 10 mK. The wide and large edge supercurrent is inaccessible for a conventional edge system and suggests the presence of non-Hermitian skin effect. A supercurrent nonlocality is also observed. The interplay between band topology and non-Hermiticity is beneficial for exploiting exotic topological matter.

The potential application of topological superconductivity in fault-tolerant quantum computing is promising[1]. Especially, the emergence of Majorana fermions at the ends of one-dimensional topological superconductors exhibits non-Abelian statistics[2], and the braiding operations on them are expected to construct topological qubits, bringing about a revolution in quantum information storage[3]. Due to the lack of natural p-wave superconductors, the idea of applying the proximity effect to materials with strong spin-orbit coupling (SOC) has been proposed[4–9]. Many efforts and attempts have been made into one-dimensional InSb (refs. 10,11) and InAs (ref. 12) semiconductor nanowires, HgTe/HgCdTe (ref. 13) and InAs/GaSb (ref. 14) quantum wells with quantum spin Hall helical edge states, and graphene quantum Hall chiral edge states[15,16], where evidence of one-dimensional supercurrent and signatures of topological nature has been revealed. Recently, the notion of topology has been extended to higher order category[17–24], opening

new routes towards the realization of Majorana zero modes on the ends of higher order hinge states when proximitized to superconductors. Evidence for one-dimensional hinge state superconductivity has been observed in Josephson junctions composed of higher order topological insulator Bi (refs. 25,26), higher order topological semimetals WTe$_2$ (refs. 27–29), MoTe$_2$ (refs. 30,31), and Cd$_3$As$_2$ (ref. 32).

However, ambiguities remain in the interpretation of super-current carried by higher order hinge states in gapless systems. Especially, the widths and amplitudes of observed edge supercurrent vary greatly in systems predicted to exhibit hinge states. In particular, the width of boundary supercurrent in topological semimetals ranges from less than 10 nm (ref. 31) to greater than 1 μm (ref. 28), and the amplitude of the boundary superconducting current varies from tens of nA (refs. 27,29,32) to several μA (ref. 28). The non-Hermitian skin effect (NHSE)[33–41] induced boundary states give an alternative edge

[1]State Key Laboratory for Mesoscopic Physics and Frontiers Science Center for Nano-optoelectronics, School of Physics, Peking University, 100871 Beijing, China. [2]Shenzhen Institute for Quantum Science and Engineering, Department of Physics, Southern University of Science and Technology, 518055 Shenzhen, China. [3]International Quantum Academy, 518048 Shenzhen, China. [4]MESA+ Institute for Nanotechnology, University of Twente, 7500 AE Enschede, The Netherlands. [5]Hefei National Laboratory, 230088 Hefei, China. ✉e-mail: anqi0112@pku.edu.cn; tanzb@sustech.edu.cn; liaozm@pku.edu.cn

supercurrent channel. In disordered crystals, the finite lifetime of quasiparticles due to electron-electron, electron-phonon, or electron-impurity scattering could give rise to non-Hermitian self-energy terms in the effective Hamiltonian[40,42–48]. Interestingly, NHSE, the piling up of eigenstates at the system's boundaries with the breakdown of traditional bulk-boundary correspondence, is shown to be universal in two and three dimensions by recent theories[41,49]. Especially, non-Hermitian skin modes, coexisting with topological hinge states, have been predicted in Dirac semimetals combined with $C_4$-symmetric perturbations[39,50,51], as illustrated in Fig. 1a (also see Supplementary Fig. 1).

In this work, we observe a broad and colossal edge supercurrent in Dirac semimetal $Cd_3As_2$-based Josephson junctions by measuring the magnetic field modulated supercurrent interference. The sample is of high carrier density, formed naturally during the synthesis process. A SQUID-like supercurrent pattern is observed, showing the edge-dominated supercurrent transport. The width and magnitude of the supercurrent edge channel are found to be ~1.6 μm and ~1 μA, which is much larger than the typical value for the topological hinge state scenario, consistent with the NHSE scenario. The skin modes are found to be boundary-sensitive. At 10 mK, the supercurrent channels at the two edges of the nanoplate are asymmetric. By increasing the temperature, the difference between the two edge channels is gradually eliminated due to thermal fluctuations. The edge supercurrent also renders a SQUID-like pattern nonlocally with a long coherence length. These observations indicate an effective mode filter of the supercurrent transport.

## Results

### SQUID-like interference pattern and edge supercurrent

Figure 1b shows the optical image of the device studied in this work. The $Cd_3As_2$ nanoplates were grown by chemical vapor deposition

method with a (112) surface plane (Supplementary Fig. 2). Individual nanoplates were proximitized with Nb superconducting contacts (see Methods). Between the Nb electrodes, a series of junctions with different channel lengths $L$ are formed. The electrical transport measurements were performed in a dilution refrigerator with a base temperature of 10 mK. Reproducible results are obtained from different junctions. The measurement results of two typical junctions (denoted by junctions 1 and 2, see Supplementary Table 1 for details) with different channel lengths are presented. The employed nanoplates are heavily electron-doped due to the native defects in the growth process[52–55], of which the Fermi level can hardly be modulated by gate voltage (Supplementary Fig. 3). The electron density is estimated to be $\sim 2 \times 10^{18}$ cm$^{-3}$ from the transfer curve (Supplementary Fig. 3d). The high electron density indicates the existence of considerable amounts of defects in the nanoplate, which would enhance the non-Hermitian perturbations and facilitate the observation of skin modes[40,42,43,45–48,56].

To investigate the spatial distribution of supercurrent, an out-of-plane perpendicular magnetic field $B_z$ is applied to induce supercurrent interference in the junctions. For the junction with a short channel length, the bulk and surface supercurrent dominate and lead to a standard Fraunhofer pattern[32]. As shown in Fig. 1c, the critical current $I_c$ of junction 2 with a channel length of 600 nm displays rapidly decaying oscillations with $B_z$ and the central lobe is almost twice as wide as the neighboring lobes, showing a typical feature of the Fraunhofer pattern. The $I_c(B_z)$ data can be well fitted using the Dynes and Fulton method (DF method)[57], as shown in Fig. 1c. The extracted supercurrent density profile (Supplementary Fig. 3e) reveals the dominant role of bulk and surface states in supercurrent transport of junction 2. When the channel length is increased, the bulk-carried supercurrents are greatly suppressed and only the supercurrent at the boundaries remains[32]. Figure 1d displays the

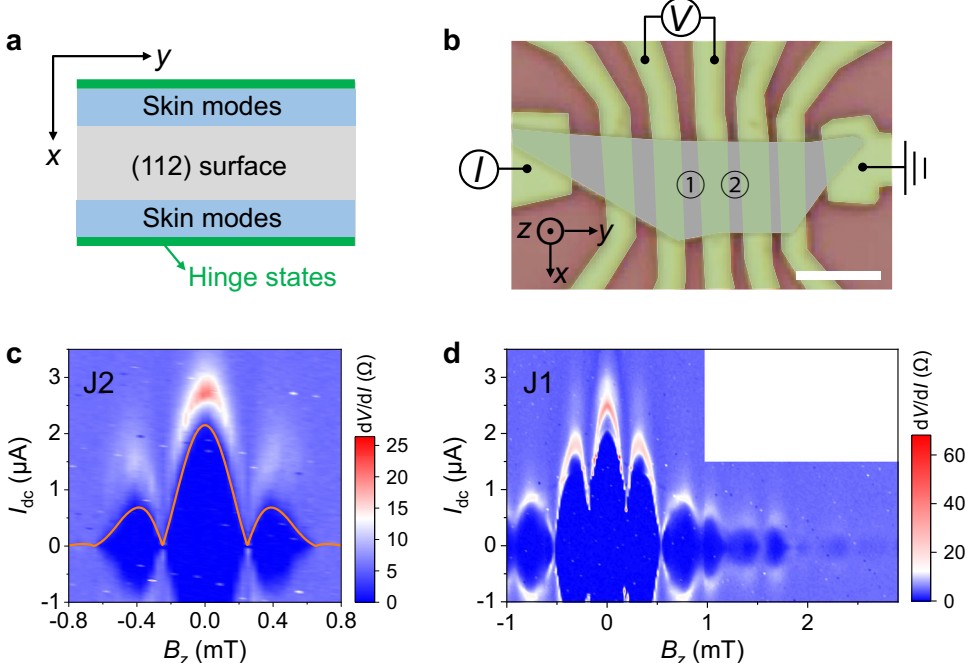

**Fig. 1 | Characteristics of the $Cd_3As_2$ nanoplate-based Josephson junctions.** **a** Hinge states (green line) and skin modes (light blue) distributed on the surface of a $Cd_3As_2$ nanoplate with (112) surface orientation. $Cd_3As_2$ has a pair of Dirac points along $k_z$ direction, i.e., (001) direction and the hinge states are the segments connecting the projection of the bulk Dirac points along the hinges. **b** Optical image of the device. The nanoplate is denoted by gray, while the Nb electrodes are denoted by green. Josephson junctions 1 and 2 are indicated, where the channel lengths are 800 and 600 nm, respectively. The four-terminal measurement method

is adopted. Scale bar, 4 μm. **c** Color map of differential resistance d$V$/d$I$ as a function of $I_{dc}$ and $B_z$ in junction 2 at the base temperature of 10 mK. The dark blue region represents the superconducting state, while its upper boundary denotes the critical current $I_c$. The orange curve shows the fitting results using the DF method. **d** d$V$/d$I$ as a function of $I_{dc}$ and $B_z$ in junction 1 at the base temperature (10 mK). A SQUID-like supercurrent interference pattern is captured, where the width of the central lobe is close to others.

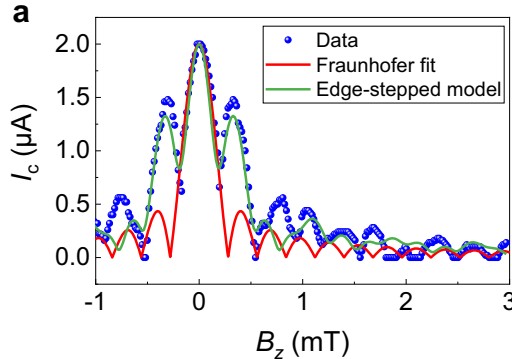

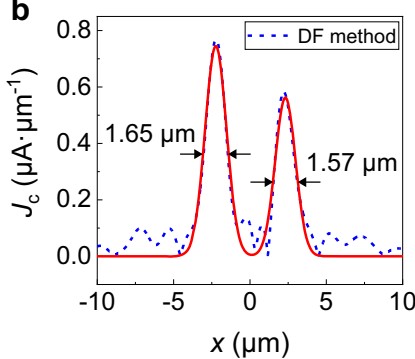

**Fig. 2 | Broad and colossal edge supercurrent observed in junction 1. a** The comparison of experimental data (scatters) extracted from Fig. 1d, Fraunhofer fit (red curve) and the edge-stepped supercurrent model fit (green curve). **b** The extracted supercurrent density profile $J_c(x)$ according to the experimental data in **a**. The blue dashed curve is obtained by the DF method, and its Gaussian fit is denoted by the red solid curve. Asymmetric supercurrent density peaks exist at two edges of the nanoplate. The full width of half maximum (FWHM) of the Gaussian peaks is 1.65 and 1.57 μm, respectively, which are considered as the width of the edge supercurrent channels.

evolution of differential resistance d$V$/d$I$ as a function of $I_{dc}$ and $B_z$ in junction 1 (channel length = 800 nm). Compared to junction 2, the $I_c$ in junction 1 can sustain at a much higher magnetic field and the widths of the central lobe and the side lobes are approximately equal. This is a typical feature of the SQUID-like pattern.

For junction 1, the data of $I_c(B_z)$ can be better fitted using the edge-stepped supercurrent model (Supplementary Fig. 4) than the Fraunhofer formula, as seen in Fig. 2a. The supercurrent density distribution $J_c(x)$ is further extracted from $I_c(B_z)$ data based on the DF method[57], as shown in Fig. 2b. The edge supercurrent is found to be dominant. Interestingly, with a Gaussian fitting (Supplementary Note 1), the width of supercurrent edge channels is found to be ~1.6 μm, much larger than the reported experimental value for other superconducting edge systems (typically several hundreds of nanometers) at low temperatures[13,14,27,29]. Also, the observed edge supercurrent is 1.18 μA for the left edge and 0.8 μA for the right edge, much larger than the value of a single hinge channel ($e\Delta_{Nb}/2\hbar = 140$ nA) in the short junction regime[27]. The non-zero $I_c$ at the first minima near the central nodes (Fig. 1d) and the two asymmetric $J_c(x)$ peaks (Fig. 2b) suggest that there are two asymmetric supercurrent channels at the two edges of the nanoplate.

**Temperature dependence**

We further investigate the temperature dependence of the edge superconductivity by increasing the temperature gradually from 10 mK to 1.6 K. The evolution of the SQUID-like pattern with temperature in junction 1 is shown in Fig. 3a. It is found that the $I_c$ of the two central nodes goes gradually from nonzero to zero with increasing temperature. Upon lifting the temperature higher than 600 mK, the zero-residual supercurrent at the central nodes suggests a symmetric edge distribution. Quantitatively, using the DF method, the calculated supercurrent density profile $J_c(x)$ at 800 mK shown in Fig. 3b indicates two symmetric edge modes. The evolution from asymmetric to symmetric edge modes is further revealed by the temperature dependence of the two-edge peak ratio, $J_c$(left peak)/$J_c$(right peak), as shown in Fig. 3c. Besides, the width of supercurrent edge channels becomes gradually equivalent for the two edges with increasing temperature (Fig. 3d).

The temperature dependence of the total critical current $I_c$ is also investigated. As shown in Fig. 3e, in the absence of a magnetic field, the $I_c$ exhibits a convex-shaped decrease with increasing $T$ up to the critical temperature, indicating so-called short junction characteristics. For a short ballistic junction, the temperature-dependent $I_c$ can be expressed by the following formula[58–60]

$$I_c(T) \propto \max_{\varphi}\left( \frac{\Delta(T)\sin(\varphi)}{\sqrt{1-\tau\sin^2\left(\frac{\varphi}{2}\right)}} \tanh\left( \frac{\Delta(T)}{2k_BT}\sqrt{1-\tau\sin^2\left(\frac{\varphi}{2}\right)} \right) \right), \quad (1)$$

where $\varphi$ is the phase difference between two superconducting electrodes, $\tau$ is the transmission coefficient of the superconductor-nanoplate interface and $\Delta(T) \sim \Delta_0\tanh(1.74\sqrt{\frac{T_c}{T}-1})$ denotes the superconducting gap of the junction and $\Delta_0$ is the proximity-induced gap at $T = 0$ K. The $\Delta_0$ and $\tau$ are treated as fitting parameters here. As shown in Fig. 3e, the experimental $I_c(T)$ data can be well-fitted by this short ballistic junction model. The transmission coefficient $\tau = 0.4$ and the induced superconducting gap $\Delta_0 = 0.26$ meV are extracted. Given that the $\Delta_0 = 1.76k_BT_c$, the obtained $T_c$ is around 1.7 K, consistent with the experimental observation.

**Nonlocal supercurrent transport on the edges**

The edge supercurrent is also found to induce a SQUID-like pattern in nonlocal setups. As sketched in Fig. 4a, a dc bias current $I_b$ (with ac excitation current $i_b$) is injected from terminals A and B, and nonlocal ac voltage difference $V_{nonlocal}$ is simultaneously monitored on the terminals C and D. The electrodes A and B form junction 1, while C and D form junction 2. When the applied $I_b$ is smaller than the critical current of junction 1 $I_c^{J1}$, the superconducting state makes the current not leak out from junction 1, rendering $V_{nonlocal} = 0$. As $I_b$ is larger than the critical current, junction 1 turns to the resistive state and the injected current $I_b$ would spread out even through the nonlocal regions, as shown in Fig. 4b. For the bulk/surface states, the current would be primarily concentrated between A and B (orange and yellow arrows in Fig. 4b), inducing a negligible nonlocal current spreading[61]. By contrast, the edge states possess a longer electron mean free path and can nonlocally propagate a longer distance, benefiting from topological protection and the suppression of backscattering[32,36,38,62]. The nonlocal current $I_{nonlocal}$ along the edge channels (blue arrows in Fig. 4b) would cause a measurable nonlocal voltage $V_{nonlocal}$ between terminals C and D, only if $I_{nonlocal}$ is larger than the critical current of junction 2 $I_c^{J2}$(ref. [61]).

Figure 4c shows the colormap of nonlocal ac voltage $V_{nonlocal}$ versus $I_b$ and out-of-plane field $B_z$. The dark blue region corresponds to the superconducting state, and the upper boundary

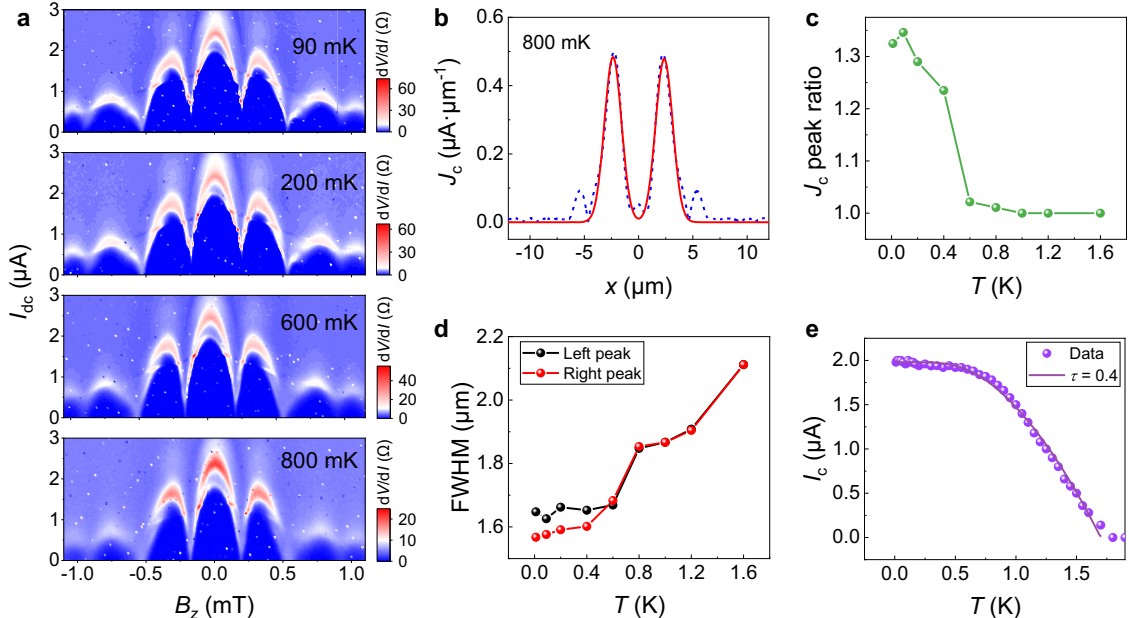

**Fig. 3 | Temperature-dependent SQUID-like interference patterns in junction 1.**
**a** Evolution of SQUID-like interference pattern with temperature. As increasing temperature, the central nodes of $I_c$ oscillations go from nonzero to zero. **b** The extracted supercurrent density profile according to the $I_c(B_z)$ data at 800 mK. The blue dashed curve is obtained by the DF method, and its Gaussian fit is denoted by the red solid curve. Symmetric $J_c$ peaks exist at two edges of the nanoplate with almost vanishing bulk contribution. **c** The ratio of $J_c$(left peak)/$J_c$(right peak) as a function of temperature. **d** The FWHM of $J_c$ edge peaks as a function of temperature, revealing the variation of the width of the supercurrent edge channel versus temperature. **e** Temperature-dependent critical current $I_c(T)$ at $B_z = 0$ mT. The purple curve shows the fitting result based on the short ballistic junction model, and the parameters of interfacial transmission coefficient $\tau = 0.4$ and proximity-induced superconducting gap $\Delta_0 = 0.26$ meV are obtained.

denotes the critical value of $I_b$ that drives the nonlocal region (junction 2) into the normal state. At $B_z = 0$ mT, $V_{nonlocal}$ is still vanishing even when $I_b$ is larger than the critical current ~2 μA of the locally biased region (junction 1), and then turns to a finite value as further increasing $I_b$ larger than 12.5 μA. In this situation, the $I_{nonlocal}$ has reached the critical current of junction 2 $I_c^{J2}$. Considering $I_c^{J2}$ ~ 2.2 μA, the $I_{nonlocal} \approx 0.17 I_b$ is thus obtained and the calculated nonlocal critical current $I_c^{nonlocal}(B_z)$ is presented in Fig. 4d. The $I_c^{nonlocal}(B_z)$ oscillations show a close resemblance to the SQUID-like pattern, which is distinct from the Fraunhofer pattern of the same region (junction 2) observed in local measurements (Fig. 1c). This nonlocal configuration acts as an effective transport mode filter, and thus the topological edge channels can be picked out. Figure 4e shows the calculated nonlocal supercurrent density profile $J_c'(x)$ in junction 2. The $J_c'(x)$ peaks are clearly observed residing on the nanoplate two edges, further confirming the supercurrent edge modes on the $Cd_3As_2$ nanoplate.

## Discussion

Here we briefly discuss the possible origins that may lead to a broad and colossal edge supercurrent. Mechanisms such as topologically trivial edge modes[63–65] and delocalization of hinge states[13], fail to explain the broadening supercurrent edge channels in our work due to coherence length and Fermi wavelength considerations (see Supplementary Note 2 for detailed discussion).

This wide and colossal supercurrent edge channel could be explained by considering the non-Hermitian skin effect[33–41]. The topological hinge states and the non-Hermitian skin modes are both present and carry the supercurrent (Fig. 1a). Therefore, the total $J_c(x)$ is much wider and larger than the predicted hinge states. To model our non-Hermitian Dirac semimetal system and demonstrate the skin effect quantitatively, we start from the effective Hermitian Hamiltonian of $Cd_3As_2$. The low-energy excitation could be described by a

minimal effective Hamiltonian[66]:

$$H_0(\mathbf{k}) = \epsilon_0(\mathbf{k}) + \begin{pmatrix} M(\mathbf{k}) & Ak_+ & 0 & 0 \\ Ak_- & -M(\mathbf{k}) & 0 & 0 \\ 0 & 0 & M(\mathbf{k}) & -Ak_- \\ 0 & 0 & -Ak_+ & -M(\mathbf{k}) \end{pmatrix}, \quad (2)$$

where $k_\pm = k_x \pm ik_y$, $\epsilon_0(\mathbf{k}) = C_0 + C_1 k_z^2 + C_2(k_x^2 + k_y^2)$ and $M(\mathbf{k}) = M_0 - M_1 k_z^2 - M_2(k_x^2 + k_y^2)$ involve the band structure. The energy dispersion follows $E(\mathbf{k}) = \epsilon_0(\mathbf{k}) \pm \sqrt{M(\mathbf{k})^2 + A^2 k_+ k_-}$, forming a pair of fourfold degenerate Dirac points at $\mathbf{k}^c = (0, 0, k_z^c = \pm\sqrt{\frac{M_0}{M_1}})$. Around the Dirac points, $M(\mathbf{k}^c) = M_0 - M_1 k_z^2 - M_2(k_x^2 + k_y^2) \approx -M_2(k_x^2 + k_y^2)$, which is in $O(k^2)$. For simplicity, we hereafter consider the expansion up to $O(k)$ and ignore terms containing $M(\mathbf{k})$. The approximation is applicable since the Fermi energy ($E_F$ ~ 80 meV, Supplementary Fig. 3) is within the linear energy dispersion range (a few hundred meV around the Dirac point)[67].

In disordered solid-state systems, the finite lifetime of quasi-particles could result in non-Hermitian terms in one-body effective Hamiltonian[40,42,43,45–48], which applies here in $Cd_3As_2$ with a high carrier density $n \sim 2 \times 10^{18}$ cm$^{-3}$ (Supplementary Fig. 3), formed naturally during the synthesis process. To illustrate the presence of non-Hermitian skin effect, we consider the non-Hermitian term of the form:

$$i\Gamma = i\gamma \mathbb{1} \otimes \sigma_x, \quad (3)$$

where $\sigma_x$ is the Pauli matrix, and $\gamma$ is a real constant reflecting the strength of the non-Hermitian interaction. The non-Hermicity is easily verified such that $\Gamma$ is a Hermitian term. Plugging the non-Hermitian part into the effective Hamiltonian of $Cd_3As_2$, the 4 × 4 quasiparticle

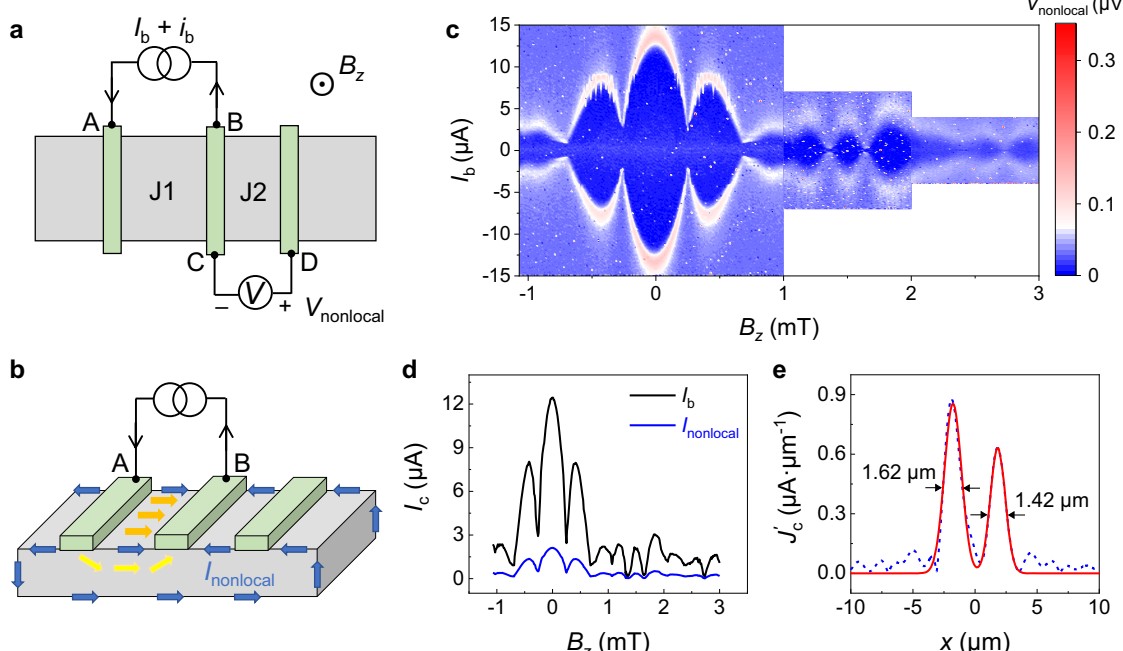

**Fig. 4 | Nonlocal measurements of supercurrent interference pattern at 10 mK.**
**a** Schematic of the nonlocal measurement configuration. The bias dc current $I_b$ superimposed on an ac excitation $i_b$ is injected from terminal A to B, and the ac voltage drop $V_{nonlocal}$ is monitored between terminals C and D. The electrodes A and B form junction 1 ($L = 800$ nm), while electrodes C and D correspond to junction 2 ($L = 600$ nm). **b** Schematic diagram of current distribution in the nonlocal measurement configuration, where the source-drain channel is in the normal state. The blue arrows represent the flowing direction of the edge current. Yellow and orange arrows denote the bulk and surface current flow, respectively. **c** The $V_{nonlocal}$ as a function of bias current $I_b$ and external field $B_z$. **d** The critical values of $I_b$ and $I_{nonlocal}$ versus magnetic field $B_z$, which drives junction 2 into the normal state and triggers a nonzero nonlocal voltage. **e** The extracted nonlocal supercurrent density distribution in junction 2. The blue dashed curve is obtained from $I_c^{nonlocal}(B_z)$ in **d**, and the red solid curve corresponds to the Gaussian fitting results.

Hamiltonian is then given by

$$H(\mathbf{k}) = H_0(\mathbf{k}) + i\Gamma$$
$$= H_0(\mathbf{k}) + i\gamma \mathbb{1} \otimes \sigma_x. \tag{4}$$

For the first two basis states, $\left|S_{\frac{1}{2}},\frac{1}{2}\right\rangle$ and $\left|P_{\frac{3}{2}},\frac{3}{2}\right\rangle$, we perform the substitution

$$k_x \rightarrow \widetilde{k_x} = k_x - i\frac{\gamma}{A}, \tag{5}$$

while for the last two basis states, $\left|S_{\frac{1}{2}},-\frac{1}{2}\right\rangle$ and $\left|P_{\frac{3}{2}},-\frac{3}{2}\right\rangle$, we do the substitution

$$k_x \rightarrow \widetilde{k_x} = k_x + i\frac{\gamma}{A}. \tag{6}$$

The Hamiltonian then writes

$$H(\widetilde{\mathbf{k}}) = \begin{pmatrix} 0 & A\widetilde{k}_+ + i\gamma & 0 & 0 \\ A\widetilde{k}_- + i\gamma & 0 & 0 & 0 \\ 0 & 0 & 0 & -A\widetilde{k}_- + i\gamma \\ 0 & 0 & -A\widetilde{k}_+ + i\gamma & 0 \end{pmatrix} + C_0$$

$$= \begin{pmatrix} 0 & Ak_+ & 0 & 0 \\ Ak_- & 0 & 0 & 0 \\ 0 & 0 & 0 & -Ak_- \\ 0 & 0 & -Ak_+ & 0 \end{pmatrix} + C_0, \tag{7}$$

which is Hermitian after the substitution. In the case of a Hermitian Hamiltonian, the Bloch states are expected. For the first two basis

states, the eigenstate evolves as

$$\psi_1 \propto e^{i\widetilde{\mathbf{k}}\cdot\vec{r}} \propto e^{i\widetilde{k_x}x} = e^{ik_x x} \cdot e^{\frac{\gamma}{A}x}, \tag{8}$$

aggregating exponentially on the right edges. For the last two basis states, the eigenstate behaves as

$$\psi_2 \propto e^{i\widetilde{\mathbf{k}}\cdot\vec{r}} \propto e^{i\widetilde{k_x}x} = e^{ik_x x} \cdot e^{-\frac{\gamma}{A}x}, \tag{9}$$

accumulating exponentially on the left edges. Thus, by modeling the Dirac semimetal Cd$_3$As$_2$ with a non-Hermitian Hamiltonian, the feature of skin modes gathering at both sides of the nanoplate along the width direction is verified.

We further take the decaying constant as the spatial broadening of the non-Hermitian skin modes, i.e., $w_{skin} = \frac{A}{\gamma}$. We adopt $A = 2.75$ eV Å obtained by fittings to the ARPES data of Cd$_3$As$_2$ (ref. 67). For the non-Hermitian amplitude, we estimate $\gamma \approx \frac{\hbar}{\tau}$, where $\tau$ is the quasiparticle lifetime. Considering the junction length of J1 in our manuscript is 800 nm, the bulk could hardly carry supercurrent due to its much smaller coherence length $\xi_{bulk} \sim 81$ nm (Supplementary Fig. 1). Therefore, we attribute the observed skin modes to the accumulation of surface states. The lifetime is then estimated to be

$$\tau \approx \frac{l_e^{surface}}{v_F} \approx 3.3 \text{ ps}, \tag{10}$$

where mean free path of surface states $l_e^{surface} \approx 1$ μm (Supplementary Fig. 1), Fermi velocity $v_F \approx 3 \times 10^5$ m/s. The non-Hermitian amplitude

follows

$$\gamma \approx \frac{\hbar}{\tau} \approx 0.20 \, \text{meV}, \tag{11}$$

giving the width of the skin modes

$$w_{\text{skin}} = \frac{A}{\gamma} \approx \frac{2.75 \, \text{eV} \, \text{Å}}{0.20 \, \text{meV}} \approx 1.4 \, \mu\text{m}, \tag{12}$$

in the same order of magnitude as our observed edge supercurrent with a width of ~1.6 μm at the base temperature of 10 mK. The consistency between the model analysis and the experimental results makes the NHSE a potential candidate for explaining the origin of wide and large edge supercurrent in our work.

We note that non-Hermitian terms are usually considered in diagonal forms to study the behaviors of quasiparticles in interacting and/or disordered many-body systems[40,42,43,45–48]. However, the condition for a diagonal self-energy matrix is not always guaranteed and needs to fulfill specific symmetry[42,44]. Considering the complexity of the multi-band electronic structure in $Cd_3As_2$, the condition of a diagonal self-energy matrix may not be satisfied. Thus, introducing an off-diagonal non-Hermitian term in an open system with disorder is a possible option, by which we successfully reveal the existence of the NHSE in Dirac semimetal Josephson junctions. The concrete physical origin of the off-diagonal non-Hermitian terms remains elusive and needs future efforts.

As for the temperature dependence, the asymmetric-symmetric transition when warming up could be understood since the thermal fluctuations would smear out the difference of the two-edge skin modes at higher temperatures. Recent theory predicts that the local pseudospectral weight, a quantitative indicator of the skin effect, no longer depends on boundary conditions at higher temperatures, due to the suppression of the line gap by a thermal perturbation[68]. Our observation of symmetric edge supercurrents that are carried by the skin mode at higher temperatures is consistent with this theoretical prediction.

In summary, we investigated the Josephson supercurrent induced into topological Dirac semimetal $Cd_3As_2$ nanoplates and we revealed that the supercurrent is carried by edge channels. Wide near-edge regions of more than 1.5 μm in width have been observed, suggesting that the quasiparticles gather near the edge, which is consistent with the theoretically predicated non-Hermitian skin effect. Through non-local measurements, the supercurrent from the bulk states was filtered out, strengthening the evidence for edge-mode supercurrents. Our work provides deep insight into the coexistence of topology and non-Hermiticity, which is significant for revealing the exotic boundary physics in topological semimetals.

## Methods
### Device fabrication
$Cd_3As_2$ nanoplates with (112) surface orientation were grown by the chemical vapor deposition method[32,69,70]. Individual $Cd_3As_2$ nanoplate was then transferred onto a silicon substrate with a 285-nm-thick $SiO_2$ coating layer. The selected nanoplate is heavily electron-doped and has a thickness of ~170 nm (Supplementary Fig. 2). Nb electrodes were fabricated after the e-beam lithography and magnetron sputtering process. The Pd capping layer was used to protect Nb from oxidization. Before the Nb deposition, an in-situ $Ar^+$ etching process was performed to remove the native oxide layer of the nanoplate.

### Transport measurements
Transport measurements were performed in a dilution refrigerator with a base temperature of 10 mK. The differential resistance was acquired by current biasing the sample with a dc component ($I_{dc}$) and

ac component ($i_{ac}$) and simultaneously measuring the ac voltage across the sample through the lock-in amplifier (SR830). For obtaining the current–voltage ($I − V$) curves, the $I_{dc}$ was swept from negative to positive and the dc voltage $V_{dc}$ was concurrently recorded. The critical current $I_c$ was determined by setting a switching threshold of $V_{dc}$. The superconducting transport measurements were conducted in the four-probe current–voltage geometry. Three-axis vector magnet was used to apply magnetic fields in different directions.

## Data availability
The data that support the findings of this study are available from the corresponding authors upon request.

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

## Acknowledgements

This work was supported by the Innovation Program for Quantum Science and Technology (2021ZD0302403 (Z.-M.L.)), National Natural Science Foundation of China (Grant Nos. 91964201 and 61825401 (Z.-M.L.)), China Postdoctoral Science Foundation (Grant No. 2021M700254 (A.-Q.W.)), and financially supported by the Netherlands Organization for Scientific Research (NWO) through a VIDI grant (VI.Vidi.203.047 (C.L.)). We are grateful to Professor Zhong Wang, Hong-Yi Wang, and Fei Song for inspiring discussions.

## Author contributions

Z.-M.L. conceived and supervised the project. C.-G.C., P.-Z.X, N.L., A.-Q.W., and Z.-C.P. grew the samples, fabricated the devices, and characterized the performance. J.-J.C. and Z.-B.T. with the guidance of D.Y. performed the transport measurements in the dilution refrigerator. Z.-M.L., A.-Q.W., C.-G.C., C.-Z.L., C.L., and A.B. analyzed the data. H.-Z.L. advised on the theoretical analysis. Z.-M.L., A.-Q.W., and C.-G.C. wrote the manuscript with discussion and input of all authors.

## Competing interests

The authors declare no competing interests.
