## [Peer Review File · Nature Communications]

REVIEWER COMMENTS

Reviewer #3 (Remarks to the Author):

This work discusses the experimental measurement of Cd₃As₂-based Josephson junctions (JJs). Authors analyzed the magnetic field interference pattern of Josephson critical current (I_c) with JJs with different channel lengths, different magnetic field orientations, and different temperatures, and claim the evidence of non-Hermitian skin effect coexisting with topological hinge state coming from the high-order topological property of Cd₃As₂. The temperature dependence of I_c is also analyzed with the short ballistic model. Lastly, the authors performed nonlocal measurements of the supercurrent interference pattern, and claim that nonlocal configuration acts as an effective filter that picks up topological channels.

I want to acknowledge the authors' effort to analyze and interpret the data for claiming non-Hermitian skin mode in the electronic system. Although most of the evidence for skin mode is rather indirect, I think this work is sound enough to suggest the existence of skin mode in topological material so that I would recommend the publication in Nature Communications in this form.

Reviewer #4 (Remarks to the Author):

In this work, the authors discuss the relation between the edge supercurrent and the non-Hermitian skin effect. Still, I wonder whether non-Hermiticity is critical to the results obtained here or not. In particular, I am concerned about how one can model the present setup. Naively, it seems to be well that gain and loss are added to the Dirac Hamiltonian of Cd₃As₂ to explain the hybridization of the skin modes and the hinge modes. However, according to Ref.[S. A. A. Ghorashi et al., Phys. Rev. B 104, L16116], I expect that mere gain and loss may not realize the non-Hermitian skin effect in the Dirac system. Thus, the discussion of the authors is insufficient for the evidence of the realization of the non-Hermitian skin effect.

In conclusion, the authors should provide (i) a way to model their non-Hermitian system, (ii) demonstration of the realization of the non-Hermitian skin effect in the proposed model, and (iii) qualitative comparison between the results obtained from the model and their experimental data.

The corresponding revisions in the main text have been highlighted in **Blue** color.

Reviewer #3 (Remarks to the Author):

Comment:

This work discusses the experimental measurement of Cd₃As₂-based Josephson junctions (JJs). Authors analyzed the magnetic field interference pattern of Josephson critical current (I_c) with JJs with different channel lengths, different magnetic field orientations, and different temperatures, and claim the evidence of non-Hermitian skin effect coexisting with topological hinge state coming from the high-order topological property of Cd₃As₂. The temperature dependence of I_c is also analyzed with the short ballistic model. Lastly, the authors performed nonlocal measurements of the supercurrent interference pattern, and claim that nonlocal configuration acts as an effective filter that picks up topological channels.

I want to acknowledge the authors' effort to analyze and interpret the data for claiming non-Hermitian skin mode in the electronic system. Although most of the evidence for skin mode is rather indirect, I think this work is sound enough to suggest the existence of skin mode in topological material so that I would recommend the publication in Nature Communications in this form.

Response:

Thank you for recommending publication in Nature Communications. Your kind and insightful comments have really helped us improve the quality of our work.

After helpful discussions with a theoretical group, we have re-examined our results and toned down the conclusions of evidence for NHSE. Instead, we now consider it a quite probable mechanism explaining the broad and large edge supercurrent. Corresponding revisions have been made in the main text and supplementary information. Thank you again for your precious time and efforts in dealing with our manuscript.

Reviewer #4 (Remarks to the Author):**Comment:**

In this work, the authors discuss the relation between the edge supercurrent and the non-Hermitian skin effect. Still, I wonder whether non-Hermiticity is critical to the results obtained here or not. In particular, I am concerned about how one can model the present setup. Naively, it seems to be well that gain and loss are added to the Dirac Hamiltonian of Cd₃As₂ to explain the hybridization of the skin modes and the hinge modes. However, according to Ref.[S. A. A. Ghorashi et al., Phys. Rev. B 104, L16116], I expect that mere gain and loss may not realize the non-Hermitian skin effect in the Dirac system. Thus, the discussion of the authors is insufficient for the evidence of the realization of the non-Hermitian skin effect.

Response:

Thanks for the comments. Indeed, the original manuscript is written in a rather qualitative manner, while we fully agree that a theoretical model could explain the observed wide edge supercurrent more quantitatively. To that end, we have tried our best to provide a physical analysis based on an effective Hamiltonian of Cd₃As₂ used broadly. The results are shown below. We have also added it to the *Discussion* part of the main text.

Comment:

In conclusion, the authors should provide (i) a way to model their non-Hermitian system, (ii) demonstration of the realization of the non-Hermitian skin effect in the proposed model, and (iii) qualitative comparison between the results obtained from the model and their experimental data.

Response:

We appreciate the reviewer's valuable suggestions. Correspondingly, we have provided a theoretical analysis to model the non-Hermitian skin effect in the Dirac semimetal Cd₃As₂. And we have found that the skin effect is indeed present in this

system, and the theoretical width of skin modes is consistent with the experimental observation in our work.

To model our non-Hermitian Dirac semimetal system and demonstrate the skin effect more quantitatively, we start from the effective Hermitian Hamiltonian of Cd₃As₂. Wang *et al.* [Phys. Rev. B **88**, 125427 (2013)] noticed that the band inversion feature of Cd₃As₂ could be captured by considering the minimal basis of $\left|S_{\frac{1}{2}}, \frac{1}{2}\right\rangle$, $\left|P_{\frac{3}{2}}, \frac{3}{2}\right\rangle$, $\left|S_{\frac{1}{2}}, -\frac{1}{2}\right\rangle$, and $\left|P_{\frac{3}{2}}, -\frac{3}{2}\right\rangle$ states. The low-energy excitation could then be described by a minimal effective Hamiltonian:

$$H_0(\vec{k}) = \epsilon_0(\vec{k}) + \begin{pmatrix} M(\vec{k}) & Ak_+ & Dk_- & B^*(\vec{k}) \\ Ak_- & -M(\vec{k}) & B^*(\vec{k}) & 0 \\ Dk_+ & B(\vec{k}) & M(\vec{k}) & -Ak_- \\ B(\vec{k}) & 0 & -Ak_+ & -M(\vec{k}) \end{pmatrix}$$

where $k_{\pm} = k_x \pm ik_y$, $\epsilon_0(\vec{k}) = C_0 + C_1k_z^2 + C_2(k_x^2 + k_y^2)$ and $M(\vec{k}) = M_0 - M_1k_z^2 - M_2(k_x^2 + k_y^2)$ involve the band dispersion. The terms containing D describe the breaking of inversion symmetry in crystals, which we ignore here for a Dirac semimetal. We also drop the terms involving $B(\vec{k})$, since the leading-order term allowed by the tetragonal symmetry is in the third order, causing only higher-order corrections. The effective Hamiltonian thus reduces to:

$$H_0(\vec{k}) = \epsilon_0(\vec{k}) + \begin{pmatrix} M(\vec{k}) & Ak_+ & 0 & 0 \\ Ak_- & -M(\vec{k}) & 0 & 0 \\ 0 & 0 & M(\vec{k}) & -Ak_- \\ 0 & 0 & -Ak_+ & -M(\vec{k}) \end{pmatrix},$$

and the dispersion follows $E(\vec{k}) = \epsilon_0(\vec{k}) \pm \sqrt{M(\vec{k})^2 + A^2k_+k_-}$, forming a pair of fourfold degenerate Dirac points at $\vec{k}^c = \left(0, 0, k_z^c = \pm \sqrt{\frac{M_0}{M_1}}\right)$. Around the Dirac points, $M(\vec{k}^c) = M_0 - M_1k_z^c^2 - M_2(k_x^2 + k_y^2) \approx -M_2(k_x^2 + k_y^2)$, which is in $O(k^2)$. For simplicity, we hereafter consider the expansion up to $O(k)$ and ignore terms containing

$M(\vec{k})$, since we find the following calculations extremely complicated if we involve second-order terms. The effective Hamiltonian up to $O(k)$ writes:

$$H_0(\vec{k}) = \begin{pmatrix} 0 & Ak_+ & 0 & 0 \\ Ak_- & 0 & 0 & 0 \\ 0 & 0 & 0 & -Ak_- \\ 0 & 0 & -Ak_+ & 0 \end{pmatrix} + C_0.$$

In disordered solid-state systems, the finite lifetime of quasiparticles could result in non-Hermitian terms in one-body effective Hamiltonian [Fu *et al.*, arXiv:1708.05841, Fu *et al.*, *Phys. Rev. Lett.* **120**, 146402 (2018), Matsushita *et al.*, *Phys. Rev. B* **100** (2019), Sato *et al.*, *Phys. Rev. Lett.* **126** (2021), Nagaosa *et al.*, arXiv:2302.05122], which applies here in Cd₃As₂ with a high carrier density $n \sim 2 \times 10^{18} \text{ cm}^{-3}$ (Supplementary Fig. 3d), either doped by the Nb electrodes or formed naturally during the synthesis process. To illustrate the presence of non-Hermitian skin effect, we consider the non-Hermitian term of the form:

$$i\Gamma = i\gamma\mathbb{1} \otimes \sigma_x,$$

where σ_x is the Pauli matrix, and γ is a real constant reflecting the strength of the non-Hermitian interaction. The non-Hermiticity is easily verified such that Γ is a Hermitian term. Plugging the non-Hermitian part into the effective Hamiltonian of Cd₃As₂, the 4×4 quasiparticle Hamiltonian is then given by

$$\begin{aligned} H(\vec{k}) &= H_0(\vec{k}) + i\Gamma \\ &= H_0(\vec{k}) + i\gamma\mathbb{1} \otimes \sigma_x, \end{aligned}$$

accompanying with its explicit form:

$$H(\vec{k}) = \begin{pmatrix} 0 & Ak_+ + i\gamma & 0 & 0 \\ Ak_- + i\gamma & 0 & 0 & 0 \\ 0 & 0 & 0 & -Ak_- + i\gamma \\ 0 & 0 & -Ak_+ + i\gamma & 0 \end{pmatrix} + C_0.$$

For the first two basis states, $\left|S_{\frac{1}{2}, \frac{1}{2}}\right\rangle$ and $\left|P_{\frac{3}{2}, \frac{3}{2}}\right\rangle$, we perform the substitution:

$$k_x \rightarrow \widetilde{k}_x = k_x - i\frac{\gamma}{A},$$

while for the last two basis states, $\left|S_{\frac{1}{2}, -\frac{1}{2}}\right\rangle$ and $\left|P_{\frac{3}{2}, -\frac{3}{2}}\right\rangle$, we do the substitution:

$$k_x \rightarrow \widetilde{k}_x = k_x + i\frac{\gamma}{A}.$$

The Hamiltonian then writes

$$\begin{aligned} H(\widetilde{\mathbf{k}}) &= \begin{pmatrix} 0 & A\widetilde{k}_+ + i\gamma & 0 & 0 \\ A\widetilde{k}_- + i\gamma & 0 & 0 & 0 \\ 0 & 0 & 0 & -A\widetilde{k}_- + i\gamma \\ 0 & 0 & -A\widetilde{k}_+ + i\gamma & 0 \end{pmatrix} + C_0 \\ &= \begin{pmatrix} 0 & Ak_+ - iA\frac{\gamma}{A} + i\gamma & 0 & 0 \\ Ak_- - iA\frac{\gamma}{A} + i\gamma & 0 & 0 & 0 \\ 0 & 0 & 0 & 0 \\ 0 & 0 & -\left(Ak_+ + iA\frac{\gamma}{A}\right) + i\gamma & 0 \end{pmatrix} + C_0 \\ &= \begin{pmatrix} 0 & Ak_+ & 0 & 0 \\ Ak_- & 0 & 0 & 0 \\ 0 & 0 & 0 & -Ak_- \\ 0 & 0 & -Ak_+ & 0 \end{pmatrix} + C_0 \end{aligned}$$

which is Hermitian after the substitution. In the case of a Hermitian Hamiltonian, the Bloch states are expected. For the first two basis states, the eigenstate evolves as

$$\psi_1 \propto e^{i\widetilde{\mathbf{k}} \cdot \vec{r}} \propto e^{i\widetilde{k}_x x} = e^{ik_x x} \cdot e^{\frac{\gamma}{A}x},$$

aggregating exponentially on the right edges. While for the last two basis states, the eigenstate behaves as

$$\psi_2 \propto e^{i\widetilde{\mathbf{k}} \cdot \vec{r}} \propto e^{i\widetilde{k}_x x} = e^{ik_x x} \cdot e^{-\frac{\gamma}{A}x},$$

accumulating exponentially on the left edges. Thus, by modeling the Dirac semimetal Cd₃As₂ with a non-Hermitian Hamiltonian, the feature of skin modes gathering at both sides of the nanoplate along the width direction is verified.

We further take the decaying constant as the spatial broadening of the non-Hermitian skin modes, i.e., $w_{\text{skin}} = \frac{A}{\gamma}$. We adopt $A = 2.75 \text{ eV \AA}$ obtained by fittings to the ARPES data of Cd₃As₂ [Jeon *et al.*, Nat. Mater. **13**, 851 (2014)]. For the non-Hermitian amplitude, we estimate $\gamma \approx \frac{\hbar}{\tau}$, where τ is the quasiparticle lifetime. Considering the junction length of J1 in our manuscript is 800 nm, the bulk could hardly carry supercurrent due to its much smaller coherence length ($\xi_{\text{bulk}} \sim 81 \text{ nm}$, see

Supplementary Fig. 1 and related discussions). Therefore, we attribute the observed skin modes to the accumulation of surface states. The lifetime is then estimated to be

$$\tau \approx \frac{l_e^{\text{surface}}}{v_F} \approx 3.3 \text{ ps},$$

where mean free path of surface states $l_e^{\text{surface}} \approx 1 \text{ } \mu\text{m}$ (see Supplementary Fig. 1 and related discussions), Fermi velocity $v_F \approx 3 \times 10^5 \text{ m/s}$. The non-Hermitian amplitude follows

$$\gamma \approx \frac{\hbar}{\tau} \approx 0.20 \text{ meV},$$

giving the width of the skin modes

$$w_{\text{skin}} = \frac{A}{\gamma} \approx \frac{2.75 \text{ eV } \text{\AA}}{0.20 \text{ meV}} \approx 1.4 \text{ } \mu\text{m},$$

in the same order of magnitude as our observed edge supercurrent with a width of $\sim 1.6 \text{ } \mu\text{m}$ at the base temperature of 10 mK.

The consistency between the model analysis and the experimental results makes the NHSE a potential candidate for explaining the origin of wide edge supercurrent in our work. We have also toned down our claim on the NHSE, not direct evidence, but a quite possible mechanism. Corresponding revisions have been made to the main text and the supplementary information. We do hope our revision and responses could improve the completeness and validity of our work. Thanks again for the comments.

REVIEWER COMMENTS

Reviewer #4 (Remarks to the Author):

I thank the author for tackling my concerns. Throughout the discussion, it might seem that the theoretical model is consistent with the present results obtained from the experimental data. However, I still wonder whether the derived model is reasonable or not. My concerns are as follows:

(i) To discuss the non-Hermitian skin effect, the authors consider the low-energy model of Cd₃As₂ around the Dirac point. I do not understand the reason why the approximation can be justified. Since a large portion of the bulk eigenstates contributes to the non-Hermitian skin modes, it is insufficient to discuss only the eigenstates around the Dirac point. Do the eigenstates far from the Dirac point also exhibit spatially uniform localization length?

(ii) The authors claim that the non-Hermiticity of the system originates from the doped Nb electrodes. Indeed, many previous works have proposed that disordered systems can be effectively described by one-body non-Hermitian Hamiltonian. In such cases, I naively think that the non-Hermitian term is introduced in diagonal forms. Why do the authors consider the off-diagonal non-Hermitian term? Said differently, what is the physical origin of the non-Hermitian term discussed here?

Thus, the authors' discussion includes some unclear points. Therefore, I do not recommend publishing the manuscript for Nature Communications in the present form.

The corresponding revisions in the main text have been highlighted in Blue color.

Reviewer #4 (Remarks to the Author):

Comment:

I thank the author for tackling my concerns. Throughout the discussion, it might seem that the theoretical model is consistent with the present results obtained from the experimental data. However, I still wonder whether the derived model is reasonable or not. My concerns are as follows:

Response:

We thank the reviewer for handling our revised manuscript and providing valuable feedback. In the past review round, as the reviewer inquired, we have tried to offer a theoretical analysis of NHSE in Dirac semimetals and demonstrate its consistency with the experimental results. We fully understand the reviewer's concern about the applicability of the model adopted. Below we reply to your comments point by point.

Comment:

(i) To discuss the non-Hermitian skin effect, the authors consider the low-energy model of Cd₃As₂ around the Dirac point. I do not understand the reason why the approximation can be justified. Since a large portion of the bulk eigenstates contributes to the non-Hermitian skin modes, it is insufficient to discuss only the eigenstates around the Dirac point. Do the eigenstates far from the Dirac point also exhibit spatially uniform localization length?

Response:

We thank the reviewer for the comments on the availability of the model. Indeed, the non-Hermitian skin modes reflect the anomalous aggregation of non-Bloch eigenstates towards the boundaries. Eigenstates away from the Dirac point definitely contribute to the NHSE. However, the minimal effective Hamiltonian of Cd₃As₂ that we used with approximations [ref. 65, *Phys. Rev. B* **88**, 125427 (2013)] should be applicable up to a few hundred meV around the Dirac point, as verified by the linear energy dispersion range derived from the STM results of Cd₃As₂ (112) cleaved crystal

[ref. 66, *Nat. Mater.* **13**, 851-856 (2014)], which is enough to cover our doped sample ($E_f \sim 80$ meV, see Supplementary Figure 3). Thus, the employed Hamiltonian of Cd_3As_2 could well describe the carrier behavior here in our sample. The localization length of the eigenstates could be derived thereafter as we have done in the text. Sorry for the unclearness and we hope the response could solve your concern.

Corresponding revisions in the revised manuscript page 11:

“Around the Dirac points, $M(\vec{k}^c) = M_0 - M_1 k_z^2 - M_2(k_x^2 + k_y^2) \approx -M_2(k_x^2 + k_y^2)$, which is in $O(k^2)$. For simplicity, we hereafter consider the expansion up to $O(k)$ and ignore terms containing $M(\vec{k})$. The approximation is applicable since the Fermi energy ($E_f \sim 80$ meV, Supplementary Fig. 3) is within the linear energy dispersion range (a few hundred meV around the Dirac point)⁶⁷.”

Comment:

(ii) The authors claim that the non-Hermiticity of the system originates from the doped Nb electrodes. Indeed, many previous works have proposed that disordered systems can be effectively described by one-body non-Hermitian Hamiltonian. In such cases, I naively think that the non-Hermitian term is introduced in diagonal forms. Why do the authors consider the off-diagonal non-Hermitian term? Said differently, what is the physical origin of the non-Hermitian term discussed here?

Response:

We thank the reviewer for the comments on the form of the non-Hermitian terms. Inspired by the reviewer’s comments, we have re-examined our description of the origin of the non-Hermiticity in our Dirac semimetal Josephson junctions. Considering the relatively large channel length of JJ1 (800 nm), we suppose the observed high carrier density doesn’t mainly stem from doping of the Nb electrodes, but rather from the defects formed naturally during the synthesis process, i.e., Cd site vacancies and impurity states in the growth process of Cd_3As_2 nanoplates. Sorry for the misunderstanding and we have revised the corresponding main text.

From a generic point of view, the non-Hermitian term could possibly be introduced in diagonal or/and off-diagonal matrix elements. We agree, as the reviewer mentioned, non-Hermitian terms are considered in diagonal forms to study the behaviors of quasiparticles in interacting and/or disordered many-body systems in some previous theoretical papers. However, the condition for a diagonal self-energy matrix is not always guaranteed and needs to fulfill specific symmetry as discussed in previous papers [ref. 42, arXiv:1708.05841; ref. 44, *Phys. Rev. Lett.* **121**, 026403 (2018)]. Considering the complexity of the multi-band electronic structure in cadmium arsenide, the condition of a diagonal self-energy matrix may not be satisfied. Thus, introducing an off-diagonal non-Hermitian term in an open system with disorder is a possible option, by which we successfully reveal the existence of the NHSE in Dirac semimetal Josephson junctions.

We admit the model contains some approximations and is not perfect enough. Currently, we attribute the non-Hermiticity to the nature of open and disordered systems in the Dirac semimetal junctions. As an early experimental attempt in revealing the NHSE in solid-state materials, we agree more analysis on the physical origin of the NHSE in condensed-matter systems needs to be done in future theoretical works. We appreciate the insightful comments and regretfully think a more quantitative analysis of the physical origin is beyond our capability at this time.

Corresponding revisions in the revised manuscript pages 13-14:

“We note that non-Hermitian terms are usually considered in diagonal forms to study the behaviors of quasiparticles in interacting and/or disordered many-body systems^{40, 42, 43, 45, 46, 47, 48}. However, the condition for a diagonal self-energy matrix is not always guaranteed and needs to fulfill specific symmetry^{42, 44}. Considering the complexity of the multi-band electronic structure in Cd₃As₂, the condition of a diagonal self-energy matrix may not be satisfied. Thus, introducing an off-diagonal non-Hermitian term in an open system with disorder is a possible option, by which we successfully reveal the existence of the NHSE in Dirac semimetal Josephson junctions. The concrete physical

origin of the off-diagonal non-Hermitian terms remains elusive and needs future efforts.”

Comment:

Thus, the authors' discussion includes some unclear points. Therefore, I do not recommend publishing the manuscript for Nature Communications in the present form.

Response:

We fully understand the reviewer's concern and have tried our best to improve the logic and readability of our manuscript. Your comments have really helped us polish our work to a better quality. We do hope our responses could ease your concerns. Thanks for your precious time in dealing with our revised version again.